# Dietary Patterns and the Risk of Inflammatory Bowel Disease: Findings from a Case-Control Study

**DOI:** 10.3390/nu13061889

**Published:** 2021-05-31

**Authors:** Reema F. Tayyem, Tamara R. Qalqili, Rawan Ajeen, Yaser M. Rayyan

**Affiliations:** 1Department of Human Nutrition, College of Health Sciences, QU Health, Qatar University, Doha 2713, Qatar; 2Biomedical and Pharmaceutical Research Unit, QU Health, Qatar University, Doha 2713, Qatar; 3Department of Nutrition and Food Technology, Faculty of Agriculture, The University of Jordan, Amman 11942, Jordan; tamara.qalqili@yahoo.com; 4Department of Nutrition, Gillings School of Global Public Health, University of North Carolina at Chapel Hill, Chapel Hill, NC 27599, USA; rajeen@email.unc.edu; 5Department of Gastroenterology & Hepatology, School of Medicine, The University of Jordan, Amman 11942, Jordan; rayyan@marshall.edu

**Keywords:** inflammatory bowel disease, ulcerative colitis, crohn’s disease, dietary patterns, Western diet

## Abstract

Scientific evidence shows that dietary patterns are associated with the risk of IBD, particularly among unhealthy and Western dietary patterns. However, Western dietary patterns are not exclusive to Western countries, as Jordanians are steadily moving towards a Western lifestyle, which includes an increased consumption of processed foods. This study aims to investigate the association between dietary patterns and the risk factors for IBD cases among Jordanian adults. This case-control study was conducted between November 2018 and December 2019 in the largest three hospitals in Jordan. Three hundred and thirty-five Jordanian adults aged between 18–68 years were enrolled in this study: one hundred and eighty-five IBD patients who were recently diagnosed with IBD (*n* = 100 for ulcerative colitis (UC) and *n* = 85 for Crohn’s disease (CD)) and 150 IBD-free controls. Participants were matched based on age and marital status. In addition, dietary data was collected from all participants using a validated food frequency questionnaire. Factor analysis and principal component analysis were used to determine the dietary patterns. Odds ratios (OR) and their 95% confidence interval (CI) were calculated using a multinomial logistic regression model. Two dietary patterns were identified among the study participants: high-vegetable and high-protein dietary patterns. There was a significantly higher risk of IBD with high-protein intake at the third (OR, CI: 2.196 (1.046–4.610)) and fourth (OR, CI: 4.391 (2.67–8.506)) quartiles in the non-adjusted model as well as the other two adjusted models. In contrast, the high-vegetable dietary pattern shows a significant protective effect on IBD in the third and fourth quartiles in all the models. Thus, a high-vegetable dietary pattern may be protective against the risk of IBD, while a high-protein dietary pattern is associated with an increased risk of IBD among a group of the Jordanian population.

## 1. Introduction

Inflammatory bowel disease (IBD) is a chronic inflammatory disease that affects the colon, ulcerative colitis (UC), or any part of the gastrointestinal tract, Crohn’s disease (CD) [1]. IBD has traditionally been a disease of the Western hemisphere. However, reports show several countries such as Japan, Hong Kong, Korea, and Eastern Europe are experiencing an increase in IBD incidence. Although not as common, an increasing incidence of IBD is identified in South Africa, South America, and Saudi Arabia [2]. Thus, the dramatic rise in the incidence of IBD, particularly in South Asia, India, and Japan (traditionally low incidence countries), suggests that environmental factors, such as a Western dietary pattern, may have an essential role in disease pathogenesis [3].

There is accumulating evidence that IBD is associated with a type of dietary pattern, micro- and macronutrients intakes, and some food items. These factors may either have therapeutic or causative implications [4]. Diet is suspected to be an environmental factor involved in the etiology of IBD [5]. Experimental models show that diet may contribute to gut inflammation through several mechanisms including antigen presentation, alteration of gut permeability, and changes in the composition of the gut microbiota [5,6,7].

Different dietary patterns are found to be associated with the risk and development of IBD. Specifically, the Western dietary pattern, which is loaded with meat, dairy products, fat, sugary foods, processed meats, pastries, sweetened drinks, alcohol, and limited amounts of vegetables and fruits was found to be associated with increasing the risk for IBD [8]. This dietary pattern contains pro-inflammatory food cytokines that can modulate intestinal permeability and alter the intestinal microbiota, promoting a low-grade chronic inflammation in the gut, which is considered an important risk factor and a prerequisite in the development of UC [9]. On the other hand, a growing body of scientific evidence indicates that a Mediterranean dietary pattern is associated with significant improvements in health status and a decrease in inflammatory markers [10,11,12]. Additionally, adherence to the Mediterranean dietary pattern shows a beneficial effect on the gut microbiome and gut metabolites (metabolome) [13,14].

In Jordan, UC and CD are not uncommon and occur among all age groups with a peak incidence in the third decade of life [15]. To our knowledge, this is the first study to investigate the association between dietary patterns and IBD risk in a selected sample of Jordanian adults. Our study aims to investigate associations between dietary patterns and the risks of UC and CD.

## 2. Materials and Methods

### 2.1. Study Design and Participants

A case-control study design was used to determine if dietary patterns are a risk factor for inflammatory bowel disease in a selected sample of Jordanian adults. In this study, 335 Jordanian adults were enrolled between November 2018 and December 2019. One hundred and eighty-five patients who were recently diagnosed with IBD (CD, *n* = 85 and UC, *n* = 100) were enrolled in this study. The number of cases and controls was decided based on two main factors: the lack of IBD prevalence in Jordan and the availability of recently diagnosed patients during the research period. Additionally, Chan et al. (2013) show that 75 participants are enough to calculate odds ratios for the risk of developing the disease [16]. One hundred and fifty IBD-free controls were recruited from the community. This includes employees and visitors from the University of Jordan Hospital, Zarqa Governmental Hospital, and Al Bashir Hospital, as well as employees working in companies and organizations. The recruited cases and controls were matched in terms of age and marital status. This study’s inclusion criteria were: adult patients between 18 to 68 years of age, patients who were recently (within three months) diagnosed with IBD, Jordanian nationality, and ability to communicate verbally and sign an informed consent form. Participants who suffer from cancers, acute appendicitis, food allergy, food intolerance, infection, primary intestinal lymphoma, intestinal tuberculosis, or anal fistulas, or are pregnant or lactating women, following therapeutic or any special diet, or unable to communicate verbally were all excluded from the study. Additionally, controls who reported symptoms of irritable bowel syndrome or anal fistulas up to one year before the start of the study were excluded.

### 2.2. Setting and Study Approval

A hospital setting was utilized for data collection. Hospitals that offer services for patients with IBD were chosen as the study sites, and primary data collection occurred in the outpatient department at each of these sites. The study and all study materials were approved by the Institutional Review Board (IRB) of each hospital (The University of Jordan Hospital IRB #22/2019-4312, Zarqa Government Hospital and Al Basheer Hospital IRB #3199). This includes obtaining approval for a private room at each study site to conduct interviews. Written informed consent was obtained from all participants prior to data collection. Information collected remained confidential, and all participants received a unique study ID. All tools and instruments used in data collection were labeled with the patient’s number. Further, all patient charts were reviewed to confirm an IBD diagnosis.

### 2.3. Data Collection

#### 2.3.1. Sociodemographic Data

Data about age, sex, marital status, working status (employee in any place or not working and retired), education (below high school, high school, diploma, bachelor, master degree, doctorate degree), smoking (smoker, nonsmoker, former-smoker), health status (suffering from any chronic diseases or not), food problems (anorexia, dysphagia, and tasteless and odorless food), current body weight, pre-diagnosis body weight, height, and waist circumferences were collected from all participants.

#### 2.3.2. Dietary Assessment

Information on diet was collected using a validated Arabic FFQ for dietary assessment, which has been previously tested for reproducibility [17]. The modified FFQ has a reasonable relative validity and reliability for energy, carbohydrate, fiber, fat, saturated fat, calcium, and iron intakes in Jordanian adults over a 1-year period. Mean energy-adjusted reliability coefficients ranged from 0.695 to 0.943. A Cronbach’s a for the total FFQ items of 0.857 was found [17]. The FFQ tracked information on the dietary history of study participants before IBD diagnosis and confirmed the control participant’s dietary habits. Standardized food models (NASCO, Saugerties, NY, USA) and standard measuring tools were used (NASCO, Saugerties, NY, USA) to determine portion size estimates. Data were collected through face-to-face interviews. Food lists in the modified FFQ questions were classified based on the type of food: 21 items of fruits and juices; 21 items of vegetables; 8 items of cereals; 9 items of milk and dairy products; 4 items of beans; 16 items of meat such as red meat (lamb and beef), chicken, fish, cold meat, and others; 4 items of soups and sauces; 5 items of drinks; 4 items of snacks and sweets; and 14 items of herbs and spices.

#### 2.3.3. Anthropometric Measurements

Participants’ body weight and height were measured using standardized techniques and calibrated tools by trained research assistants. Body weight was measured to the nearest 0.1 kg with minimal clothing and without shoes using a calibrated scale (Omron, Japan). Height was measured to the nearest cm when participants were in the full standing position without shoes using a calibrated measuring rod [18]. BMI was calculated by dividing weight in kilograms by the square of height in meters [18]. A trained dietitian carried out all anthropometric measurements.

Seven-day Physical Activity Recall (PAR): Seven-Day PAR is a questionnaire that focuses on a participant’s recall of the usual time spent doing a physical activity over a seven-day period [19]. The validated PAR questionnaire was used to measure physical activity level [19].

### 2.4. Statistical Analysis

All statistical analyses were conducted using SPSS version 22.0 (IBM SPSS Statistics for Windows, IBM Corporation). The significance level was set at *p* ≤ 0.05. For descriptive statistics, mean ± standard deviation (SD) and percentages were used. *t*-tests evaluated the differences between cases and controls in continuous variables, and Chi-square was used to detect differences among categorical variables. ANOVA was used with Fisher’s least significant difference (LSD) method to find the differences between the three groups; controls and UC and CD patients. Dietary patterns were derived using factor analysis with principal component analysis (PCA) method. Consumption frequency was used to identify the dietary patterns. The foods in the FFQ were separated into 21 food items based on their similarity of nutrient content and culinary usage or their reported relationship with IBD (Table 1). Table 1 shows the factor-loading matrix for the 2 retained factors. These factors explained 45.6% of the total variance in the original data set. The rotation had the effect of making loadings either all positive or all negative for each factor. Only the magnitude of each loading was used to name the factors. A Kaiser–Meyer–Olkin (KMO) test and Bartlett’s test of sphericity were used to assess suitability for using factor analysis for this exercise. Sampling adequacy and inter-correlation of factors were supported by KMO value > 0.657 and Bartlett’s test of sphericity < 0.001, respectively. Factors were retained based on an eigenvalue of >1.25 for the screen plot. Further, Varimax rotation was applied to review the correlations between variables and factors. Food items with absolute factor loadings > 0.35 were considered to have contributed significantly to the pattern. Cases and controls received an individual factor score for each identified pattern. Potential confounders (adjusted for age, gender, BMI, smoking, marital status, total energy intake, education level, and physical activity) were chosen based on reported risk factors for IBD [20]. Odds ratios (OR) and their 95% confidence interval (CI) were calculated using a multinomial logistic regression model.

## 3. Results

Table 1 shows that the first factor, the high-vegetable dietary pattern, has the greatest loading in colored pepper, mixed vegetables, fresh tomato, onion, and olive pickles. The second factor, the high-protein dietary pattern, has the greatest loadings on beef mortadella, burger, canned tuna, chicken mortadella, chicken, liver, and egg.

Three hundred and thirty-five Jordanian adults aged between 18–68 years (185 were recently diagnosed with IBD (UC: *n* = 100 (35.0% male); CD: *n* = 85 (44.7% male); and IBD-free controls; *n* = 150 (49.3% male)) were recruited in this case-control study. Table 2 shows no significant differences in marital status, work status, smoking, and educational level among cases and controls. However, there is a significant difference between controls and IBD cases in the means of current body weight, BMI, and waist circumference (*p* < 0.05). Additionally, there is a significant difference in physical activity between UC and controls and CD and controls.

The ORs and corresponding 95% CI of IBD cases and controls by quartiles of factor scores and continuous factor scores for the two dietary patterns are shown in Table 3. Results refer to the composite model, including the two dietary patterns together and the relevant confounding variables. There is a significant increase in the risk of IBD with a high-protein intake at the third and fourth quartiles for the crude (OR, CI: 2.196 (1.046–4.610); 4.391 (2.67–8.506), respectively), model I adjusted pattern (OR, CI: 2.216 (1.146–4.210); 4.215 (2.166–8.204), respectively), and model II adjusted pattern (OR, CI: 2.196 (1.101–4.357); 5.452 (2.646–11.232), respectively). On the other hand, a high-vegetable dietary pattern shows a significant protective association for IBD on the third and fourth quartiles in the three models (Table 3).

## 4. Discussion

The key result of this study is consistent with prior studies for the presence of an association between IBD and dietary patterns. This case-control study’s findings shed light on the relationship between IBD and dietary patterns among Jordanian adults who were recently diagnosed with IBD.

Dietary patterns are defined as the quantity, variety, or combination of different foods and beverages in a diet and the frequency at which they are habitually consumed. Previous research reports that several different dietary patterns exist among individuals with IBD, including healthy dietary patterns, dietary patterns high in vegetables and fruits, Western dietary patterns, Mediterranean dietary patterns, high-protein dietary patterns, and high-fat dietary patterns [21]. This study focuses on exploring two of those identified dietary patterns: high-vegetable and high-protein. A high-vegetable dietary pattern has the greatest loading in colored pepper, mixed vegetables, fresh tomato, onion, and olive pickles. Meanwhile, a high-protein dietary pattern has the greatest loadings on beef mortadella, burger, canned tuna, chicken mortadella, chicken, chicken liver, and egg.

Our study shows that a high-vegetable dietary pattern has a significant protective association with IBD in the third and fourth quartiles of the three dietary models. Limdi et al. (2018) found that phytochemicals (lignans, flavonoids, and antioxidants) in fruit, cereals, and vegetables exerted anti-inflammatory effects through growth factors, maintenance of intestinal barrier integrity, and an antioxidant effect [22]. Additionally, several studies demonstrate that a high-fruit-and-vegetable dietary pattern might positively influence IBD risk due to the abundance of antioxidant properties [23,24,25]. Researchers have also found that high-fruit-and-vegetable intake is associated with a decreased risk of IBD. These findings were replicated in our study [23,24,25]. The authors found that the association between fruit and vegetable intake significantly decreased the risk of IBD. Amre et al. (2007) showed that higher amounts of vegetables (OR 0.69, 95% CI 0.33–1.44, *p* = 0.03), fruits (OR 0.49, 95% CI 0.25–0.96, *p* = 0.02), fish (OR 0.46, 95% CI 0.20–1.06, *p* = 0.02), and dietary fiber (OR 0.12, 95% CI 0.04–0.37, *p* < 0.001) protected from CD [25].

Regarding the second dietary pattern of this study’s focus, high-protein intake, we found a significantly increased risk of IBD at the third and fourth quartiles for the crude and the two adjusted patterns. These findings align with reports made by several other studies that indicate that a high total protein intake (from animal sources) is associated with an increased risk of IBD [26,27,28]. Peters et al. (2021) reveal that a dietary pattern loaded by grain products, oils, potatoes, processed meat, red meat, condiments and sauces, and sugar, cakes, and confectionery can exacerbate the flares of IBD significantly (HR: 1.51, 95% CI: 1.04–2.18, *p* = 0.029) during follow-up [26]. Similarly, a meta-analysis of case-control studies by Hou et al. (2011) shows a positive correlation between animal protein consumption and whole protein intake and CD [28]. In the same context, a study performed by Jantchou et al. (2010) shows that a high total protein intake, specifically animal protein, is associated with a significantly increased IBD risk [27]. However, Jantchou et al. (2010) also find no significant positive association between high vegetable protein and an increase in IBD risk [27]. In a study by Jowett et al. (2004), they found that protein metabolites provide substrates for gut bacteria, which impacts microbiome composition or short-chain fatty acids (SCFA) and further affects enterocyte function [29]. A postulated mechanism set forth by Blachier et al. (2019) reports that protein intake can exacerbate IBD by its sulfur and high cysteine content, which is utilized by sulfate-reducing bacteria to generate hydrogen sulfide (H_2_S) that in turn has detrimental inflammatory effects [30]. Lastly, Vidal-Lletjós et al. (2017) stated in their review that an excessive intake of protein may increase the intestinal production of potentially toxic bacterial metabolites [31]. These metabolites may affect the epithelial repair process through inhibiting the colonic epithelial cell respiration, cell proliferation, and/or affecting barrier function [31].

In addition, diet plays a pivotal role in the clinical care of patients with IBD and CD. For digestive system disorders, nutritional therapy can be implemented as a primary or supplemental treatment option for IBD and CD. Furthermore, diet and nutritional therapy play a major role in disease management for IBD/CD patients by correcting malnutrition and micro- and macro-nutrient deficiencies, reversing metabolic pathological consequences of disease and nutrient deficiency, increasing oral intake of nutritional supplements, and providing vital and structured recommendations on specific dietary patterns to better manage symptoms and further limit damage to the digestive system. As highlighted in our study, it is important to consider focusing on evident food properties and food groups that work to improve the overall nutritional status of IBD/CD patients while delivering optimal clinical care and diet-centered disease management. Similarly, additional studies emphasize the impact that dietary intervention can have regarding nutritional adequacy in disease management. Cioffi et al. (2020), in a cross-sectional study of 117 adult patients with CD in Italy, reveals that CD patients have a low fiber intake presented by a low-residue diet to reduce diarrhea or avoid abdominal pain. Therefore, assessment of nutritional intake can be critical in promoting dietary intervention and focused nutritional counseling to improve nutritional status for IBD/CD patients [32].

This study is novel because it is the first study conducted in Jordan to highlight the association between dietary patterns and IBD risk. Therefore, our study has several notable strengths. Those strengths include the use of a prospective study design and restriction of the study population to only include recently diagnosed cases over the past three months. These methods ameliorate the potential for selection bias traditionally seen in a retrospective study design. Second, using a validated Arabic FFQ that was modified to reflect food consumption patterns in Arab countries, especially in Jordan, ensures effectively capturing cultural variance in our study. Additionally, our use of standardized food models and measuring tools to estimate portion sizes strengthens our study measures by ensuring accurate and reliable data collection. Lastly, we confirmed all CD and UC cases through a thorough medical record review and face-to-face questionnaire process, a significant advantage over studies that rely on discharge codes.

Our study has limitations. Limitations include that our dietary questionnaire recalled dietary patterns in IBD patients for only one year, which is considered a short duration to significantly impact the pathophysiology of IBD. However, we believe that the recall period of one year used in this study is very likely reflective of the previous years, as most participants indicated a consistent dietary pattern in the past five years. Second, our study cohort may not be representative of the overall Jordanian population. To address this concern, we included three large representative areas in the country as our study sites. We chose three large referral hospitals that serve many patients from urban and rural areas in the country.

## 5. Conclusions

In conclusion, this study provides substantial evidence supporting the presence of an association between dietary patterns and the risk of IBD. Overall, two dietary patterns, high-vegetable intake and high-protein intake, were identified in this study as having a role in IBD risk. A significantly increased risk of IBD was detected with high-protein intake at the third quartile for the crude and the two adjusted models (OR, CI: 2.196 (1.046–4.610); 2.212 (1.051–4.654); 2.580 (1.087–6.122), respectively). In contrast, high-vegetable dietary patterns show a significant protective association with IBD in all three models’ quartiles.

## Figures and Tables

**Table 1 nutrients-13-01889-t001:** Factor loadings for the two rotated factors in IBD (*n* = 185) and controls (*n* = 150).

Food Items	Dietary Patterns	
	High-Vegetables	High-Protein
Beef Mortadella		0.842
Burger		0.877
Canned Tuna		0.874
Chicken Mortadella		0.648
Chicken		0.724
Chicken Liver		0.724
Egg		0.724
Cauliflower	0.471	
Colored Pepper	0.890	
Mixed Vegetables	0.890	
Fresh Tomato	0.890	
Green Beans	0.431	
Carrot	0.454	
Onion	0.890	
Peas	0.462	
Labaneh		0.423
Milk	−0.360	
White Cheese		0.423
Olive Oil	−0.393	
Sunflower Oil	−0.393	
Olive Pickles	0.890	
% Variance Explained	24.33	21.27

**Table 2 nutrients-13-01889-t002:** Socio-demographic and anthropometric measurements of the study participants.

Variables	Participants					*p*-Value
	Control (*n* = 150)	IBD Total	*p*-Value	UC (*n* = 100)	CD (*n* = 85)	
*N* (%)						
Gender						
Male	74 (49.3)	73 (39.5)	0.244	35 (35)	38 (44.7)	0.081
Female	76 (50.7)	112 (60.5)		65 (65)	47 (55.3)	
Marital Status						
Married	116 (77.3)	141 (78.4)	0.493	75 (75)	66 (77.6)	0.928
Single	25 (15.1)	28 (15.7)		17 (17)	11 (12.9)	
Divorce	6 (6.4)	11 (3.2)		5 (5)	6 (7.1)	
Widow	3 (3.2)	5 (2.7)		3 (3)	2 (2.4)	
Education Level						
Below the high school	7 (4.7)	9 (4.9)	0.205	5 (5)	4 (4.7)	0.546
High school	46 (30.7)	70 (37.8)		37 (37)	33 (38.8)	
Diploma	26 (17.3)	37 (20.0)		16 (16)	21 (24.7)	
Bachelor	59 (39.3)	57 (30.8)		36 (36)	21 (24.7)	
Master degree	9 (6)	8 (4.3)		3 (3)	5 (5.9)	
Doctorate degree	3 (2)	4 (2.2)		3 (3)	1 (1.2)	
Work Status						
Employee	96 (64)	84 (45.4)	0.596	46 (46)	38 (44.7)	0.003
Not employed or retired	54 (36)	101 (54.6)		54 (54)	47 (55.3)	
Food Problem						
Yes (anorexia, dysphagia, and tasteless and odorless food)	7 (4.7)	20 (10.8)	0.258	14 (14)	6 (7.05)	0.139
No	143 (95.3)	165 (89.2)		86 (86)	79 (92.9)	
Family History						
Yes	-	37 (20.0)	0.001	24 (24)	13 (15.3)	0.001
No	150 (100)	148 (80.0)		76 (76)	72 (84.7)	
Cigarette Smoking						
Smoker	57 (38)	62 (33.5)	0.074	25 (25)	37 (43.5)	0.001
Non-smoker	91 (60.7)	99 (63.8)		60 (60)	39 (45.9)	
Former-smoker	2 (1.3)	24 (2.7)		15 (15)	9 (10.6)	
Duration of Suffering from IBD						
Duration Less than 30 days	-	75 (40.5)	-	29 (29)	46 (54.1)	0.001
Duration from 30–60 days	-	61 (33.0)		39 (39)	22 (25.9)	
Duration from 60–90 days	-	49 (26.5)		32 (32)	17 (20)	
Mean ± SD						
Age (years)	41.4 ± 12.5	39.8 ± 12.6	0.446	39.8 ± 11.9	41.2 ± 12.8	0.580
Height (Cm)	166.4 ± 0.1 ^a^	164.8 ± 8.3	0.128	162.5 ± 0.1 ^b^	168.4 ± 0.1 ^a^	0.001
Current Weight (Kg) (Measured)	74.8 ± 13.1 ^a^	71.0 ± 13.0	0.007	66.9 ± 11.9 ^b^	75.6 ± 12.8 ^a^	0.001
Previous weight (Kg) (Self-reported)	74.3 ± 12.7 ^a^	72.7 ± 13.4	0.460	68.6 ± 12.5 ^b^	77.5 ± 13.1 ^a^	0.001
Current BMI (kg/m^2^)	27.1 ± 4.4 ^a^	27.0 ± 4.7	0.050	25.5 ± 4.9 ^b^	26.8 ± 4.9 ^a^	0.021
Waist Circumference (cm)	89.04 ± 9.0 ^a^	88.4 ± 9.5	0.019	83.9 ± 9.3 ^b^	89.9 ± 13.0 ^a^	0.001
Physical activity (Met/week)	2479.4 ± 296.8 ^a^	19018 ± 673.8	0.011	1971.4 ± 887.6 ^b^	1818.8 ± 887.2 ^b^	0.033

*p*-value ≤ 0.05 considered significant. *t*-test was used to find the difference between controls and IBD cases, while ANOVA was used with Fisher’s LSD to find the differences between the three groups: controls and UC and CD patients. Different two letters (a and b) means that there is a significant difference between three variables.

**Table 3 nutrients-13-01889-t003:** Association between IBD risk and dietary patterns among the IBD (*n* = 185) and controls (*n* = 150).

Variables	Q1	Q2	Q3	Q4
Crude OR and CI ^#^				
High-Vegetable	1	0.648 (0.339–1.24)	0.136 (0.068–0.271)	0.126 (0.064–0.248)
Controls/Cases	21/68	20/60	51/32	59/27
High-Protein	1	1.130(0.572–2.235)	2.196 (1.046–4.610)	4.391 (2.67–8.506)
Controls/Cases	43/40	57/27	25/46	25/52
Adjusted OR and CI Model 1 *				
High-Vegetable	1	0.669 (0.241–1.2853)	0.142 (0.071–0.285)	0.128 (0.064–0.255)
Controls/Cases	21/68	20/60	51/32	59/27
High-Protein	1	1.091(0.549–2.165)	2.216 (1.146–4.210)	4.215 (2.166–8.204)
Controls/Cases	43/40	57/27	25/46	25/52
Adjusted OR and CI Model 2 **				
High-Vegetable	1	0.664 (0.336–1.2853)	0.127 (0.060–0.261)	0.114 (0.055–0.237)
Controls/Cases	21/68	20/60	51/32	59/27
High-Protein	1	1.003(0.483–2.085)	2.196 (1.101–4.357)	5.452 (2.646–11.232)
Controls/Cases	43/40	57/27	25/46	25/52

^#^ OR and CI: odd ratio and confidence interval. * Adjusted for age and gender. ** Adjusted for age, BMI, gender, physical activity, energy, marital status, education level, and smoking.

## Data Availability

The datasets analyzed and/or generated during the current study are available from the corresponding author on reasonable request.

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
