# Peer review of "Dietary Patterns and the Risk of Inflammatory Bowel Disease: Findings from a Case-Control Study"

_nutrients, 2021, doi:10.3390/nu13061889_

Round 1
Reviewer 1 Report
The article is well structured and written. However, i recommend 3 small changes:
- In the statistical analysis, it should be indicated whether the variables were tested to assess the normality of the distribution and which test was used to make this assessment.
- The methodology should indicate what is considered high vegetables intake and high protein intake. The cutoff points for each quartile should be identified in the table with regard to the intake of protein and vegetables.
-In the discussion (lines 37 to 42) the indicated phrases must be removed and placed in the results because the phrases indicate results that are not part of the main objective of the study. In conclusion, the authors should focus on responding to the objectives proposed in the article.
Author Response
Dear Editor:
We are pleased to resubmit the revised version for publication and truly we are very thankful to you for reviewing our manuscript thoroughly. We have revised the present research paper in light of the useful suggestions and comments from you. All the corrections have been highlighted in yellow color at the manuscript's document. All the manuscript content has been revised critically by a native English speaker.
We hope our responses to the comments are up to the level of satisfaction.
With many thanks.
The authors
Responses to comments
Reviewer 1:
Major
1) In the statistical analysis, it should be indicated whether the variables were tested to assess the normality of the distribution and which test was used to make this assessment.
Response: It has been added.
2) The methodology should indicate what is considered high vegetables intake and high protein intake. The cutoff points for each quartile should be identified in the table with regard to the intake of protein and vegetables.
Response: Using Principal Component Analysis (PCA) is based mainly on the frequency of food consumption. Therefore, we won't be able to indicate cutoff points of the quartiles of the vegetable intake.
3) In the discussion (lines 37 to 42) the indicated phrases must be removed and placed in the results because the phrases indicate results that are not part of the main objective of the study. In conclusion, the authors should focus on responding to the objectives proposed in the article.
Response: It has been removed

Reviewer 2 Report
I read with great interest this study, aiming to evaluate the role of dietary patterns in the subsequent development of IBD in Jordanian population, where the IBD incidence is increasing in the last years. By performing a case-control study, the Authors enrolled 185 newly diagnosed IBD patients which were compared with a matched cohort of 150 non-IBD controls. They found that a high-fruit and vegetable dietary pattern was protective against IBD development; by contrast, a high-protein dietary pattern was associated with an increased risk of IBD.
Although these associations have been explored in other studies, this study highlights the importance of such dietary patterns in the development of IBD in countries with a previous low IBD incidence, increasing our knowledge about IBD pathogenesis.
The study is well-written and well conducted, the manuscript is clear.
I have some comments which I believe could improve the manuscript.
- Table 2 might be confusing, since the Authors presented data for controls, UC and CD, while the other tables compared controls and IBD (all). It is not clear which differences are statistically significant (controls vs CD? Controls vs UC? Controls vs CD vs UC?). Therefore, I would like to add another column in the Table 2, namely IBD. The Authors could compare control and IBD (in general) by reporting specific p values for each variable, then the comparison between controls and UC and CD.
- It could be very interesting to evaluate whether specific dietary patterns differ between UC and CD. The Authors could add a further table (similar to table 3 – but comparing controls with UC and with CD) in order to show differences (if any).
- The Discussion section should be increased. In particular, the Authors should underline the fact that diet plays a pivotal role not only in the pathogenesis of IBD, as demonstrated in the present study, but also in the management of the disease. In effect, recent studies demonstrate the importance of nutritional evaluation in IBD as it predicts the outcome of patients (see Cioffi et al, Marra et al). The assessment of dietary intake can be crucial for optimizing dietary intervention with focused nutrition counseling, to improve nutritional status in IBD patients.
Please, added these concepts in the Discussion section and the suggested references ( - Cioffi I, et al. Assessment of bioelectrical phase angle as a predictor of nutritional status in patients with Crohn's disease: A cross sectional study. Clin Nutr. 2020; - Marra M, et al. New Predictive Equations for Estimating Resting Energy Expenditure in Adults With Crohn's Disease. PEN J Parenter Enteral Nutr. 2020; - Cioffi I, et al. Evaluation of nutritional adequacy in adult patients with Crohn's disease: a cross-sectional study. Eur J Nutr. 2020)
Author Response
Dear Editor:
We are pleased to resubmit the revised version for publication and truly we are very thankful to you for reviewing our manuscript thoroughly. We have revised the present research paper in the light of the useful suggestions and comments from you. All the corrections have been highlighted in yellow color at the manuscript's document. All the manuscript content has been revised critically by native English speaker.
We hope our responses to the comments are up to the level of satisfaction.
With many thanks.
The authors
Responses to comments
Reviewer 2:
- Table 2 might be confusing, since the Authors presented data for controls, UC and CD, while the other tables compared controls and IBD (all). It is not clear which differences are statistically significant (controls vs CD? Controls vs UC? Controls vs CD vs UC?). Therefore, I would like to add another column in the Table 2, namely IBD. The Authors could compare control and IBD (in general) by reporting specific p values for each variable, then the comparison between controls and UC and CD.
Response: It has been added.
- It could be very interesting to evaluate whether specific dietary patterns differ between UC and CD. The Authors could add a further table (similar to table 3 – but comparing controls with UC and with CD) in order to show differences (if any).
Response: Many thanks for this suggestion. We already analyzed the dietary patterns of the UC and CD separately but because the sample size of CD, the results were unreadable. Therefore, combining both CD and UC was a must.
3. The Discussion section should be increased. In particular, the Authors should underline the fact that diet plays a pivotal role not only in the pathogenesis of IBD, as demonstrated in the present study, but also in the management of the disease. In effect, recent studies demonstrate the importance of nutritional evaluation in IBD as it predicts the outcome of patients (see Cioffi et al, Marra et al). The assessment of dietary intake can be crucial for optimizing dietary intervention with focused nutrition counseling, to improve nutritional status in IBD patients.
Please, added these concepts in the Discussion section and the suggested references ( - Cioffi I, et al. Assessment of bioelectrical phase angle as a predictor of nutritional status in patients with Crohn's disease: A cross sectional study. Clin Nutr. 2020; - Marra M, et al. New Predictive Equations for Estimating Resting Energy Expenditure in Adults With Crohn's Disease. PEN J Parenter Enteral Nutr. 2020; - Cioffi I, et al. Evaluation of nutritional adequacy in adult patients with Crohn's disease: a cross-sectional study. Eur J Nutr. 2020)
Response: Cioffi I, et al. study "Evaluation of nutritional adequacy in adult patients with Crohn's disease: a cross-sectional study. Eur J Nutr. 2020" has been added to the discussion. The other two references are not relevant to the objective of our study.

Reviewer 3 Report
The study by Reema Tayyem et al. reports on dietary factors that are found associated with the risk of IBD. This is a well-written paper in a timely subject. However, I have some comments regarding some aspects of the study.
- Did the authors have validated their dietary data regarding volunteers consumption with biological indicators, like for instance uremia as an indicator of protein intake?
- Some important references are missing regarding the association between high-protein diet consumption and risk of IBD, as well as regarding the way high-protein diet can influence the course of inflammatory bowel diseases (see notably Shoda et al; Am J clin Nutr 1996; Spooren et al. Aliment Pharmacol Ther 2018; Vidal-Lletjos et al; Nutrients 2017; Steck et al; Gut 2012). These aspects are insufficiently discussed in regards with the knowledge on that topic.
- The authors should be aware that hydroben sulfide, depending on its concentration, may act either as an anti-inflammatory factor or as a pro-inflammatory factor (see the recent review by Blachier et al. Current Opin Clin Nutr Metab Care 2019).
Author Response
Dear Editor:
We are pleased to resubmit the second revised version for publication and truly we are very thankful to you for reviewing our manuscript thoroughly. We have revised the present research paper in light of the useful suggestions and comments from you. All the corrections have been highlighted in blue color at the manuscript's document. All the manuscript content has been revised critically by a native English speaker.
We hope our responses to the comments are up to the level of satisfaction.
With many thanks.
The authors
Responses to comments
Reviewer 3:
- Did the authors have validated their dietary data regarding volunteers consumption with biological indicators, like for instance uremia as an indicator of protein intake?
Response: The used FFQ was validated using another dietary assessment but not biochemical parameter. The paper of validating the FFQ had been published in 2014 at Journal of Academy of Nutrition and Dietetics
Tayyem RF, et al. (2014), Validation of a Food Frequency Questionnaire to assess macronutrient and micronutrient intake among Jordanians. Journal of Academy of Nutrition and Dietetics. 114 (7):1046-1052
2. Some important references are missing regarding the association between high-protein diet consumption and risk of IBD, as well as regarding the way high-protein diet can influence the course of inflammatory bowel diseases (see notably Shoda et al; Am J clin Nutr 1996; Spooren et al. Aliment Pharmacol Ther 2018; Vidal-Lletjos et al; Nutrients 2017; Steck et al; Gut 2012). These aspects are insufficiently discussed in regards with the knowledge on that topic.
Response:
Vidal-Lletjos et al; Nutrients 2017: has been added to the discussion. Thanks for your suggestion.
Spooren et al. Aliment Pharmacol Ther 2018: more updated reference has been added for the same authors.
Steck et al; Gut 2012: This reference did not discuss the high protein intake. The main objective of the review is to highlight the relevance of host- and bacteria-derived proteases and their signaling mechanisms.
3. The authors should be aware that hydroben sulfide, depending on its concentration, may act either as an anti-inflammatory factor or as a pro-inflammatory factor (see the recent review by Blachier et al. Current Opin Clin Nutr Metab Care 2019).
Response: Blachier F, Beaumont M, Kim E, Cysteine-derived hydrogen sulfide and gut health, Current Opinion in Clinical Nutrition and Metabolic Care: January 2019 - 22 - 1 - 68-75
This reference has been added

Round 2
Reviewer 2 Report
The manuscript has been improved after revision
Author Response
Thank you so much.
No more comments are needed.
Reviewer 3 Report
The authors had adequately taken my comments into consideration. Some editing of the text is required but can be done on the proofs of the paper (for instance line 214 intake not ntake).